# Drainage-structuring of ancestral variation and a common functional pathway shape limited genomic convergence in natural high- and low-predation guppies

James R. Whiting[1]*, Josephine R. Paris[1], Mijke J. van der Zee[1], Paul J. Parsons[1,2], Detlef Weigel[3], Bonnie A. Fraser[1]

1 Department of Biosciences, University of Exeter, Exeter, United Kingdom, 2 Department of Animal and Plant Sciences, University of Sheffield, Sheffield, United Kingdom, 3 Department of Molecular Biology, Max Planck Institute for Developmental Biology, Tübingen, Germany

* jwhiting2315@gmail.com

## Abstract

Studies of convergence in wild populations have been instrumental in understanding adaptation by providing strong evidence for natural selection. At the genetic level, we are beginning to appreciate that the re-use of the same genes in adaptation occurs through different mechanisms and can be constrained by underlying trait architectures and demographic characteristics of natural populations. Here, we explore these processes in naturally adapted high- (HP) and low-predation (LP) populations of the Trinidadian guppy, *Poecilia reticulata*. As a model for phenotypic change this system provided some of the earliest evidence of rapid and repeatable evolution in vertebrates; the genetic basis of which has yet to be studied at the whole-genome level. We collected whole-genome sequencing data from ten populations (176 individuals) representing five independent HP-LP river pairs across the three main drainages in Northern Trinidad. We evaluate population structure, uncovering several LP bottlenecks and variable between-river introgression that can lead to constraints on the sharing of adaptive variation between populations. Consequently, we found limited selection on common genes or loci across all drainages. Using a pathway type analysis, however, we find evidence of repeated selection on different genes involved in cadherin signaling. Finally, we found a large repeatedly selected haplotype on chromosome 20 in three rivers from the same drainage. Taken together, despite limited sharing of adaptive variation among rivers, we found evidence of convergent evolution associated with HP-LP environments in pathways across divergent drainages and at a previously unreported candidate haplotype within a drainage.

## Author summary

Convergent evolution is the process whereby similar phenotypes evolve in response to common selection in independent lineages, providing strong evidence of adaptation in

**Data Availability Statement:** Raw sequencing reads are available on ENA: PRJEB43917 (Aripo, Madamas, Tacarigua) and PRJEB10680 (Guanapo

and Oropouche). Final VCF data are available on FigShare, doi: 10.6084/m9.figshare.14315771. Other data and scripts used to analyse data are available on github: github.com/JimWhiting91/guppy_convergence. This repository is archived with Zenodo, doi: 10.5281/zenodo.4740381.

**Funding:** JRW, PJP, and BAF are funded by the H2020 European Research Council (ERC; erc. europa.eu) (GUPPYCon 758382). JRP and BAF are funded by the Natural Environment Research Council (NERC; nerc.ukri.org) (NE/P013074/1). DW is funded by the Max-Planck-Gesellschaft (www.mpg.de/en) (WEIGEL). This project utilised equipment funded by the Wellcome Trust (www. wellcome.org) Institutional Strategic Support Fund (WT097835MF), Wellcome Trust Multi User Equipment Award (WT101650MA) and Biotechnology and Biological Sciences Research Council (BBSRC; bbsrc.ukri.org) LOLA award (BB/K003240/1). The funders had no role in study design, data collection and analysis, decision to publish, or preparation of the manuscript.

**Competing interests:** The authors have declared that no competing interests exist.

response to natural selection. This process can involve changes at the same regions of the genome, known as genomic convergence. We explore this in the replicated evolution of high- and low-predation Trinidadian guppies, an important model system for studies of phenotypic evolution, but where little is known about the underlying genetics. Our findings highlight that limitations on how genetic variation is distributed have restricted the same mutations or genes being involved in the convergent evolution of high- and low-predation guppies, but different genes of similar function are likely involved. We also highlight and examine a large candidate region associated with three rivers from the same drainage. Our results demonstrate constraints on genomic convergence at certain levels, but suggest there is some repeatability in the genetic basis of convergent phenotypic evolution in this important model system. Genomic convergence in the guppy system is therefore more limited than in other prominent study systems, suggesting the pervasiveness of this process in nature is highly context-dependent.

## Introduction

The process of adaptation in nature can be thought of as a complex interplay between random happenstance and repeatable processes in independent lineages. The latter of these, often termed convergent or parallel evolution [1,2], has provided a myriad of examples from which general rules and principles of adaptation have been dissected under natural conditions. Empirical evidence accumulated over the last decade has demonstrated that convergent phenotypes are often encoded by convergent changes at the genetic level across many taxa (reviewed in [3–6]). We are now at the point where we can ask why genetic convergence ranges from common in some systems to non-existent in others.

While, there are many definitions of genetic convergence (or parallelism) [2], here we use it broadly to describe selection acting at any of three levels: on the same mutations (eg. [7–9]); different mutations affecting the same genes (eg. [10–12]); or different genes in the same functional pathways (eg. [13–16]). Further, variation among lineages may arise through one of three modes: either *de novo* mutations (eg. [17,18]); as shared ancestral variation (eg. [19,20]); or through introgression among lineages (eg. [21–24]). An emerging trend within systems is that adaptation involving multiple traits can involve combinations of these levels and modes of convergence. For example, stickleback adapting to freshwater experience selection on ancestral *eda* haplotypes [25] and *de novo* mutation at the *pitx1* gene [10] to repeatedly evolve freshwater bony armour plate and pelvis phenotypes respectively. Similarly, Pease et al. [26] found all three modes of convergence occurring across a clade of wild tomato accessions: adaptive introgression of alleles associated with immunity to fungal pathogens, selection on an ancestral allele conferring fruit colour, and repeated *de novo* mutation of alleles associated with seasonality and heavy metal tolerance. Further, the same phenotype may arise in response to the same selection through any of the above modes, as observed in glyphosate-resistance amaranths across North America [27]. In this study, the authors found glyphosate-resistance evolved in one location by introgression and selection on a pre-adapted allele, in another by the fixation of a shared ancestral haplotype, and in a third location through selection on multiple, derived haplotypes.

Given the differences in modes and levels of genetic convergence observed across empirical studies, various contingent factors have emerged as important. These include the redundancy in the mapping of genotype to phenotype [28,29], i.e. how many genetic routes exist to replicate phenotypes? For example, simple one-to-one mapping is expected to result in reuse of the

same genes or even mutations, while redundancy can lead to convergent phenotypes by selection on different genes in shared functional pathways. In addition, population structure dictates the sharing of adaptive variation among lineages by which selection may act on [30]. Finally, two lineages may experience an aspect of their environment in a similar way, but in a multidimensional sense environmental variation may limit genetic convergence through pleiotropic constraint [31]. This may result in the re-use of genes with minimal constraint and minimal effects on other aspects of fitness, as suggested for *MC1R* in pigmentation across vertebrates [32]. Alternatively, similarity of environments within multivariate space can predict genetic convergence [33–35], whereby consistencies in the multidimensional fitness landscape channel adaptation along conserved paths. Conversely, inconsistencies may offer up alternative routes to fitness peaks. This combination of pleiotropy and differences among fitness landscapes may also explain why genetic convergence can vary for the same traits in the same species in global comparisons, for example in comparisons of Pacific-derived vs Atlantic-derived freshwater stickleback [35–37].

It is clear then to understand the complexity by which genetically convergent evolution might emerge requires study systems for which we already have abundant research on interactions between phenotype and environment. Here, we make use of a model of phenotypic convergence, the Trinidadian Guppy (*Poecilia reticulata*), to evaluate genetic convergence in the replicated adaptation of low-predation (LP) phenotypes from high-predation (HP) sources. For approaching 50 years, this system has provided valuable insights into phenotypic evolution in natural populations, including some of the first evidence of rapid phenotypic evolution in vertebrates across ecological, rather than evolutionary, timescales [38,39]. The guppy has since become a prominent model of phenotypic evolution in nature, but accompanying genomic work has only recently begun to emerge.

The topography of Northern Trinidad creates rivers punctuated by waterfalls, which restrict the movement of guppy predators upstream but not guppies themselves. This replicated downstream/HP and upstream/LP habitat has produced convergent HP-LP guppy phenotypes; LP guppies produce fewer, larger, offspring per brood [40,41], differ in shoaling behaviour [42,43], swimming performance [44] and predator evasion [45], and exhibit brighter sexual ornamentation [46]. However, whilst LP guppies evolve brighter colouration, the repeatability of specific colour patterns is limited and can be non-parallel [47,48]. Rearing second generation HP-LP guppies in a laboratory setting with controlled rearing conditions confirms that observed differences in life history have a genetic basis [49], and additional work has further demonstrated heritability for colour [50,51] and behaviour [52]. Alongside studies of natural populations, the convergent and replicated nature of these phenotypes has been established with experimental transplanting of HP populations into previously uninhabited LP environments, in which LP phenotypes evolve in only a few generations [38,53–55].

Here, we examine whole-genome sequencing of five replicated HP-LP population pairs (S1 Table) across the main drainages of Northern Trinidad: Caroni (Tacarigua, Guanapo, Aripo rivers), Northern (Madamas river) and Oropouche (Oropouche/Quare river). Importantly, none of the sites, including the Aripo, are introduction experiments. Previous work looking at HP-LP convergence in natural HP-LP guppy populations using reduced representation RAD-sequencing found some evidence of molecular convergence [56]. This study however only included three natural populations pairs and inferences from RAD-sequencing can be limited by reliance on linkage disequilibrium and an inability to pinpoint specific candidate genes. To comprehensively explore genetic convergence in this system we first examine how genetic variation is distributed across Northern Trinidad by quantifying population structure, between-river introgression, and within-river demography. We then compare and contrast selection scans between HP-LP pairs within each river to detect signals of convergent

evolution. Finally, we examine a large candidate haplotype to explain the mode and mechanisms by which convergence may have occurred at this genomic region.

## Results

### Population structure, admixture, and demographic history

Prior to assessing genomic convergence, it is important to contextualise neutral processes such as population structure, introgression and past demography. In doing so, we establish expectations regarding how potentially adaptive genetic variation is distributed and shared among populations; informing on the most likely mode by which genetic convergence may occur in this system.

SNAPP (SNP and AFLP Package for Phylogenetic analysis; Figs 1B and S1) [57], and fineSTRUCTURE (Fig 1C) [58] confirmed that each river's HP-LP pair formed sister

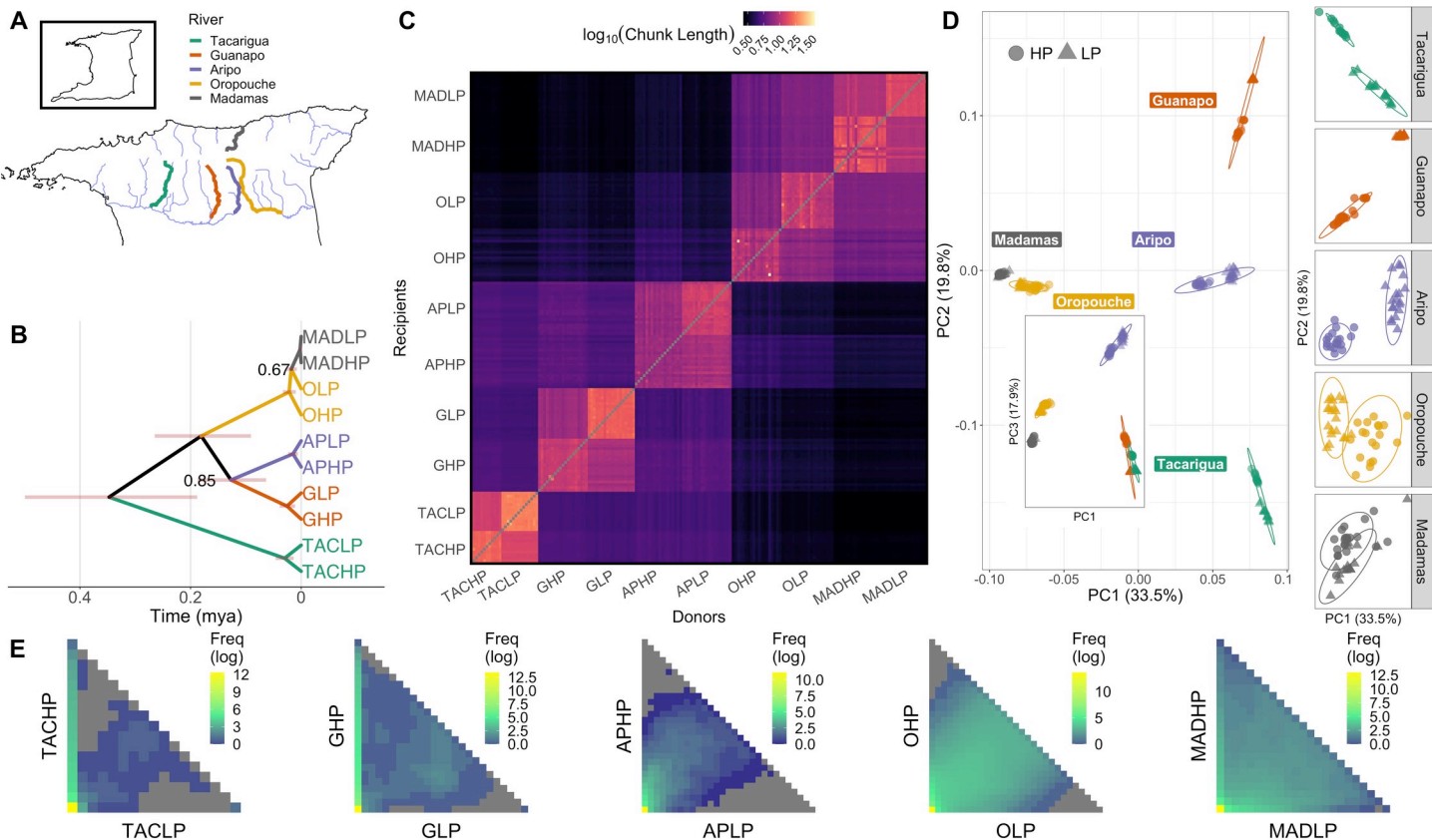

**Fig 1. Sampling sites, population structure and admixture across the five rivers.** Map (A) highlights sampled rivers from the three drainages in Northern Trinidad. Other major rivers are illustrated in light blue. Sampling rivers plotted alongside topography is available in S3 Fig. The coastline shapefile was sourced from OpenStreetMap (openstreetmap.org; CC BY-SA 2.0), rivers were added manually and are available as shapefiles at Zenodo doi: 10.5281/zenodo.4740381. The dated phylogeny (B) of sampling populations based on SNAPP, highlighting sister statuses of each HP-LP pair. Red regions around nodes denote 95% uncertainty around dates. Nodes with <90% support are indicated with their posterior support. Heatmap (C) illustrates the length of painted recipient haplotypes based on donor haplotypes, used by fineSTRUCTURE to infer coancestry (values are log10-transformed). PCA (D) with populations coloured according to river and shaped according to predation regime. PC1 and PC3 are presented in the insert. In the main panel, ellipses denote river clusters at 95% confidence intervals. The right-hand column shows zoomed-in regions around each river cluster on PC1 and PC2, with ellipses highlighting separation and 95% confidence intervals of predation clusters. Heat maps in (E) show projected two-dimensional site frequency spectra (2dsfs) for each river pair highlighting the sharing of variants between LP and HP populations within each river. LP populations are on the x-axis and HP populations are on the y-axis. In each sfs, the frequency of sites in each population is illustrated from 0 to 2 N, where N is the number of individuals in each population. Each cell within these 2dsfs therefore shows the density (log-transformed) of SNPs with relevant allele counts in each population. Cells within the first column and first row show private alleles that are absent in one population (allele count of 0). Grey cells are missing data, where no SNPs are found at allele counts of x and y in LP and HP populations respectively.

populations and structure between pairs (as river units) was well-defined. The dated phylogeny, based on divergence with the outgroup *P. wingei* (3.41 mya)[59]), suggested that HP-LP population splits were generally of the order of thousands of years old (S2 Table). Madamas was estimated as the youngest (1999 years, 95% HPD interval 2–4617), whilst Tacarigua was the oldest (29922 years, 95% HPD interval 13717–46331 years). These figures should be treated cautiously however due to differing rates of contemporary migration between HP-LP populations (see below).

Expectedly, the strongest population structure separated rivers by drainage, with PC1 (33.5%) separating rivers from the Caroni drainage (Guanapo [GHP,GLP], Aripo [APHP, APLP], Tacarigua [TACHP,TACLP]) from the Northern and Oropouche drainages (Madamas [MADHP, MADLP] and Oropouche [OHP, OLP] rivers, respectively) (Fig 1D). This split was dated at approximately 0.18 mya (S2 Table), which is considerably more recent than previous estimates of the divergence between Caroni and Oropouche drainages based on mitochondrial phylogenies [60]. PC2 (19.8%) separated out Caroni rivers, highlighting population structure within this drainage is stronger than structure between Madamas and Oropouche, despite these rivers being in separate drainages. The shared node of all Caroni rivers was dated at approximately 0.34 mya (S2 Table). These PCA axes were robust to the removal of individual rivers from the Caroni drainage, demonstrating the stronger population structure in the Caroni drainage is not a sampling artefact (S2 Fig). In all cases (with the exception of Madamas, where HP-LP cluster separation is minimal), LP populations were found to be further from the global centroid in PCA space relative to their respective HP populations (Fig 1D). This pattern demonstrates elevated drift from the common ancestral state, and is strongly suggestive that LP populations are derived from HP sources.

Admixture proportions inferred by fineSTRUCTURE agreed with stronger structure within the Caroni drainage, and were lower on average (based on shared haplotype lengths and number of chunk counts) between rivers within the Caroni drainage than between Oropouche and Madamas (Fig 1C). We also detected signatures of introgression between APHP and the Oropouche/Madamas lineage, categorised as elevated haplotype donor proportions of individuals from these rivers into APHP recipients (Fig 1C).

To quantify genome-wide introgression, we calculated D and Fbranch ($f_b$) statistics for all trios with *Dsuite* (version 0.3; [61]) (Fig 2). Briefly, D statistics reveal ABBA-BABA imbalances within trios and an outgroup (*P. wingei* in this case, see methods), which may be due to either introgression between non-sisters, or incomplete lineage sorting. The $f_b$ statistic is a heuristic approach to summarise f4-admixture ratios from across the whole tree topology to identify introgression between specific nodes/tips. We observed statistically significant (bonferroni-corrected p-value < 0.001) D-statistics across many trios, however these often exhibited minimal f4-ratios. Conservatively, we focused then on trios where P1 and P2 were true sisters, i.e. HP and LP populations from the same river (S3 Table), with significant D statistics and f4-ratio > 0.05 and supporting evidence from the $f_b$ summary.

The strongest signal of introgression was observed between the upstream LP sites OLP and MADLP (Fig 2; $f_b$ = 0.204, Z = 29.12), suggesting introgression has most likely occurred between these populations (also producing signatures of introgression between OLP and MADHP). Additional cross-drainage introgression was observed between APHP-OHP ($f_b$ = 0.058, Z = 14.85) and APHP-OLP ($f_b$ = 0.052, Z = 16.15). This indicates introgression has taken place between downstream Aripo and a population from the neighbouring Oropouche river, which are around 70m apart at the height of the wet season [62]. We also observed introgression between upstream APLP and the other Caroni drainage populations (TACHP, GHP, GLP), that was strongest between APLP and GHP ($f_b$ = 0.140, Z = 24.90). The $f_b$ summary also highlighted excessive allele sharing among some lineages, namely between the Guanapo/Aripo

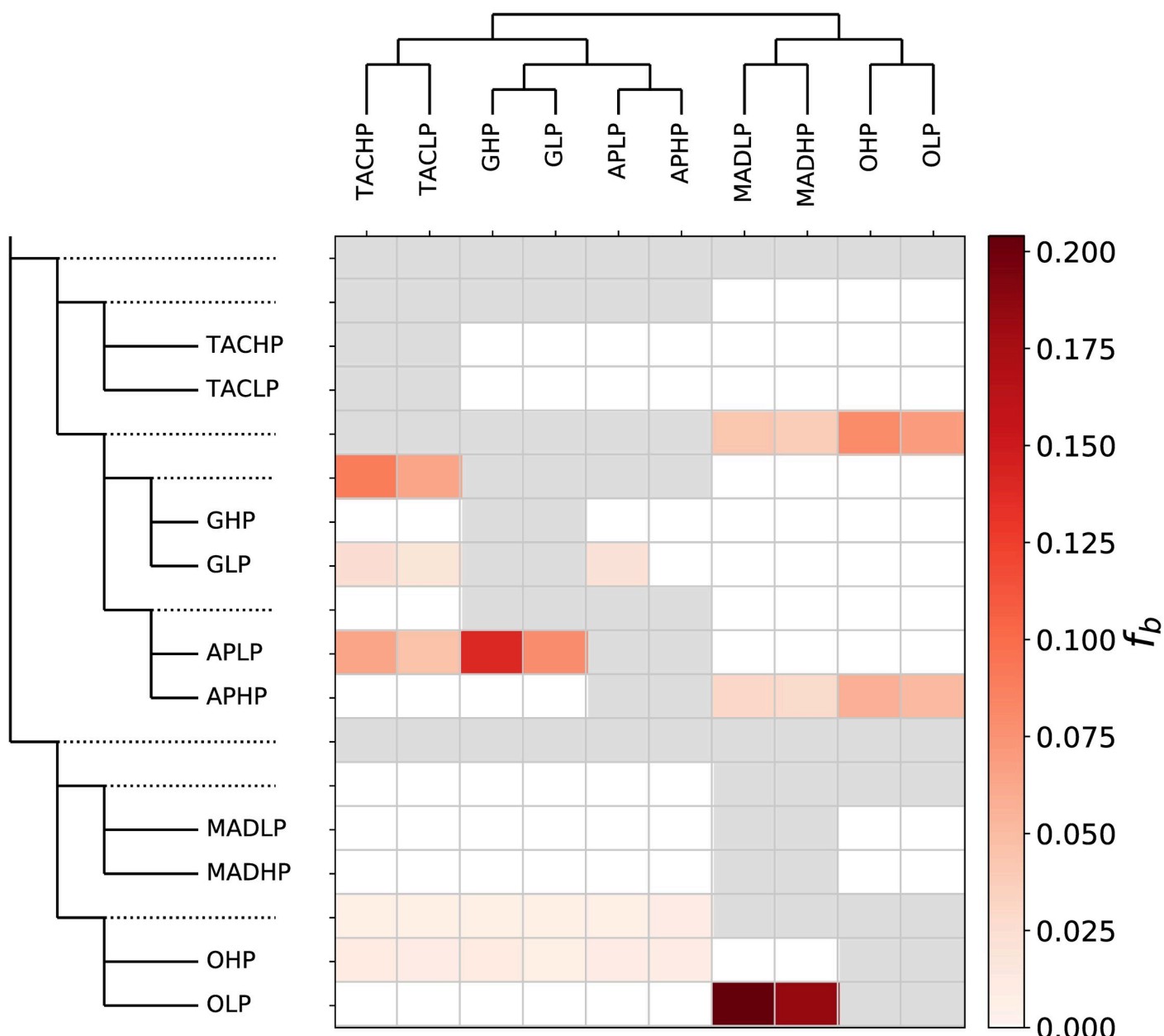

**Fig 2. Fbranch ($f_b$) summary of introgression among the ten sampled populations.** Rows represent nodes within the tree topology, and columns represent tips. Each cell shows the $f_b$ statistic between a tree node (rows) and each tip (column). Grey cells are empty where comparisons cannot be made.

lineage and the Madamas/Oropouche lineage, and between the Guanapo lineage and Tacarigua lineage.

To assess within river population demography (i.e. LP population bottlenecks, HP-LP migration), we performed demographic modelling based on two-dimensional site frequency spectra (2dSFS) using *fastsimcoal2* [63](Fig 1E and Table 1). All demographic models performed better with the addition of migration, which in every case was higher downstream from LP to HP. For three rivers (Aripo, Madamas, Tacarigua) a historic LP bottleneck was detected alongside stable current population size. Guanapo was better supported by a model with no HP population growth, and Oropouche by a model that suggested HP population

**Table 1. Demographic parameters of each river, inferred by *fastsimcoal2*.** Values in brackets represent confidence intervals of 95% after bootstrapping 100 SFS.

| Drainage | River | Mean HP-LP $F_{ST}$ | HP $N_e$ | LP $N_e$ | HP > LP Migration | LP > HP Migration | Model |
|---|---|---|---|---|---|---|---|
| Caroni | Tacarigua | 0.243 | 3,549 (2864–4493) | 474 (347–612) | 4.78E-05 | 1.55E-04 | LP Bottleneck |
| Caroni | Guanapo | 0.269 | 19,698 (19592–19698) | 155 (135–183) | 3.88E-06 | 8.34E-04 | No population changes |
| Caroni | Aripo | 0.072 | 43,354 (33767–59439) | 6,122 (4588–9008) | 2.10E-04 | 3.27E-04 | LP Bottleneck |
| Oropouche | Oropouche | 0.087 | 4,514 (4148–4910) | 3,740 (3825–4411) | 2.43E-04 | 1.21E-03 | HP population growth |
| Northern | Madamas | 0.171 | 1,121 (922–1355) | 2,480 (2337–2743) | 8.68E-05 | 4.88E-04 | LP Bottleneck |

growth. The particularly high estimates of $N_e$ in APHP agree with the above analyses of introgression into this population.

Altogether, these analyses illustrate how genetic variation is segregated across the five rivers in our dataset. Primarily, ancestral variation is dictated by geography, with populations defined within rivers, then within drainages. Particularly strong population structuring is observed in the Caroni drainage (Tacarigua, Guanapo and Aripo rivers), with limited introgression having occurred among these rivers within drainage. In contrast, we detect significant introgression across drainages, particularly between the Madamas and Oropouche rivers, demonstrating the potential for shared genetic variation among rivers. Gene flow among rivers, particularly among upstream regions, is likely facilitated by flooding events, although we expect this to be more difficult within the Caroni drainage due to steeper mountain topography between rivers (S3 Fig). The modest introgression observed within the Caroni drainage may occur through physical connectivity of the rivers (Fig 1A).

Within rivers, we see evidence of population bottlenecks in LP populations, potentially limiting the amount of available adaptive variation. This is particularly apparent in Tacarigua and Guanapo, within which LP populations have particularly low $N_e$ estimates, an excess of monomorphic sites that are polymorphic in the HP founder, and only limited private polymorphic sites (Fig 1E). In other words, the variation within these LP populations is a subset of that found in the corresponding HP. For all river pairs, our demographic modelling agrees that migration upstream from HP to LP is weaker than LP to HP, compounding the potential for limited variation upstream. Some HP-LP populations are better connected by migration however, such as Oropouche and Madamas, where many polymorphic sites are shared between upstream LP and downstream HP. Altogether, this amounts to predictable constraints and limitations on the sharing of adaptive variation among LP populations, which is observed in the pairwise 2dsfs among LP populations (S4 Fig), with the exception of OLP and MADLP due to introgression.

## Candidate HP-LP regions and assessing convergence

To evaluate regions associated with HP-LP adaptation, we scanned the genome using several approaches: XtX [64], a Bayesian analogue of $F_{ST}$ that includes a simulated distribution under neutrality; AFD, absolute allele frequency difference, which scales linearly from 0–1 between undifferentiated and fully differentiated [65]; and XP-EHH (extended haplotype homozygosity) [66], which compares homozygosity between phased haplotypes between populations. To identify selected regions, we calculated each measure in non-overlapping 10kb windows within each river between HP and LP sites. Putatively selected windows were identified if they were detected as outliers by at least two approaches (see methods for outlier criteria for individual tests; S5 Fig). Using an intersect of all three may be over-conservative. For example, we would miss instances where divergent selection within a river fixes alternate haplotypes, such that both HP and LP populations have similarly low heterozygosity (i.e. no XP-EHH outlier but an

outlier in XtX and AFD). Typically, windows were identified that had either high XtX and high AFD, or high XP-EHH and high AFD (S4 Table), although reasonable overlap among all three selection scans was also observed. Overlapping selection scan outlier windows contained more SNPs on average than a genome-wide expectation (S6A Fig).

Comparing the intersecting list of candidates of XtX, AFD and XP-EHH within each river revealed little overlap among rivers, with only a single 10kb window overlapping in more than two rivers (Fig 3A; for genome-wide plots see S7 and S8 and S9 Figs). We then scanned the genome further with 100kb sliding windows (50kb increments) to assess potential clustering of outlier windows in larger regions, but this approach similarly revealed little overlap among rivers. We then explored whether outlier regions (10kb windows overlapping in >1 selection scans) were enriched for genes in common biological pathways between rivers using one-to-one zebrafish orthologues, which may suggest repeated pathway modification through different genes. Using the outlier regions defined above, no pathway was significantly overrepresented in any river. We did however notice cases in which the same pathways exhibited fold-enrichments >1 in multiple rivers (Fig 3B), albeit non-significant within rivers in each case.

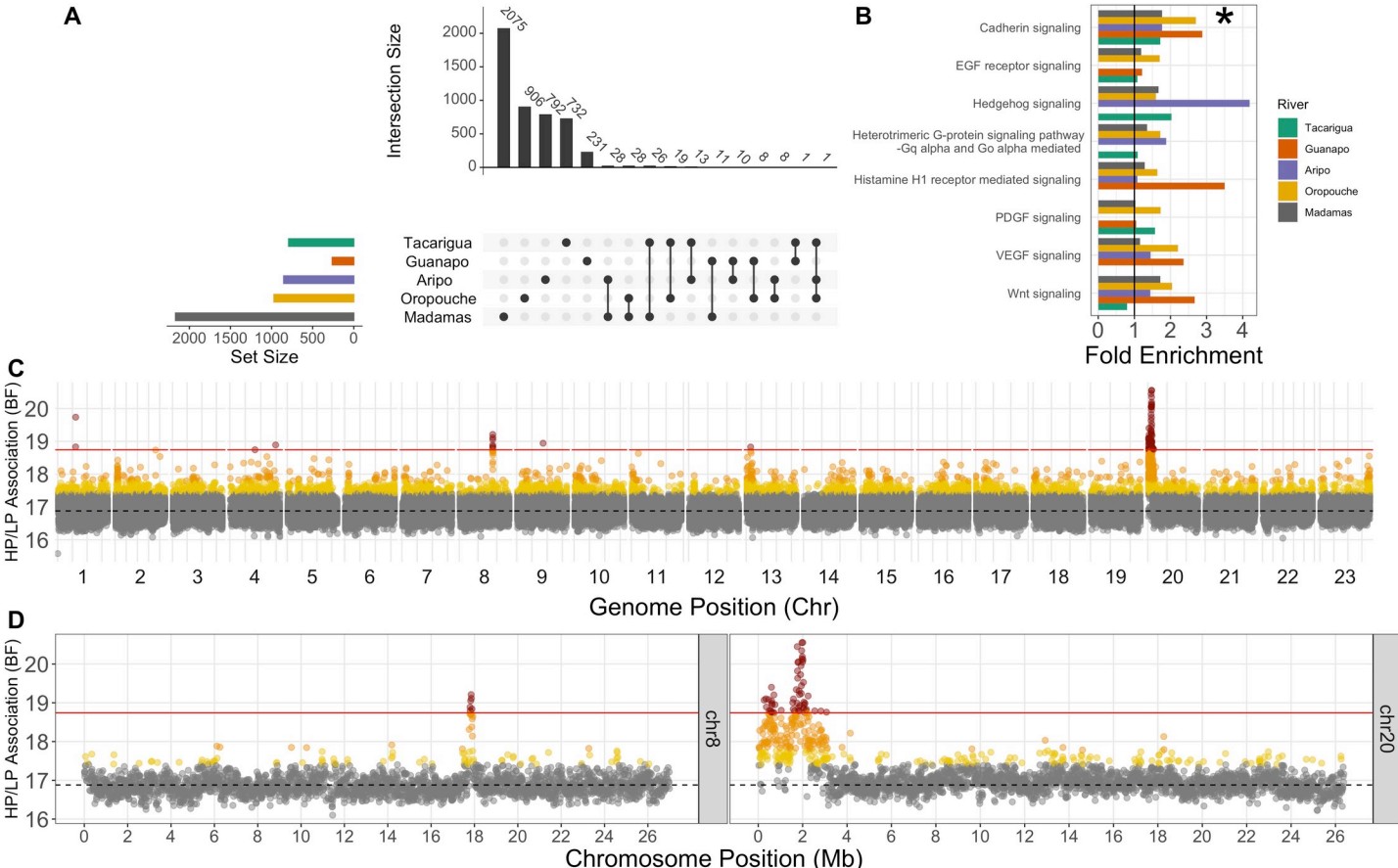

**Fig 3. Selection scan results and evidence of convergence.** Upset plot (A) of overlap among outlier sets (evidence from two of AFD, XtX, XP-EHH) highlighting no overlap beyond sets of three rivers. Coloured horizontal bars show the total set size (number of outlier windows) detected in each river. Vertical bars denote the size of the overlap among sets, equivalent to an overlapping region within a Venn diagram. The overlapping region is described by filled points below vertical bars. Fold enrichment of pathways (B) associated with zebrafish orthologs of genes in outlier regions within each river calculated from the Panther DB. Cadherin-signaling pathway was the only pathway with fold-enrichment >1 in all rivers. Other presented pathways had fold-enrichment >1 in four rivers. Genome-wide BayPass scan results (C) scanning 10kb windows for association with HP-LP classification. Points are coloured according to quantiles: 95% = yellow, 99% = orange, 99.9% = red. Dashed line represents the median BF, and the solid red line denotes 99.9% quantile cut-off. Peaks on chromosomes 8 and 20 are also highlighted (D).

We used permutations to explore the likelihood of observing fold-enrichment >1 in our five independently-derived outlier sets. This analysis identified that Cadherin-signaling pathway genes were overrepresented across all five rivers relative to by-chance expectations ($p$ = 0.013) (Fig 3B). In total, 20 genes from the Cadherin-signaling pathway were recovered from all five river outlier sets, with some overlap between them (S5 Table). This analysis may be over-conservative, due to analysing only guppy genes with one-to-one zebrafish orthologues. Other genes associated with cadherin-signaling were detected by our selection scans, including *Cadherin-1* and *B-Cadherin* in a differentiated region on chromosome 15 (~ 5 Mb) in Oropouche and Tacarigua, but these genes exhibited a many-to-many orthology with zebrafish genes so were omitted. We also examined pathways with fold-enrichment >1 in any four rivers, but these were not significant ($p > 0.05$) according to permutation tests (Fig 3B).

We next associated allele frequency changes with HP-LP status using BayPass' auxiliary covariate model. This latter approach has the advantage of using all populations together in a single analysis, whilst controlling for genetic covariance. As above, BayPass outlier windows contained more SNPs on average than a genome-wide expectation (S6B Fig). Scans for regions associated with HP-LP classification identified two major clusters of associated 10kb windows on chromosomes 8 and 20 (Fig 3C and 3D). In total, we highlighted 70 10kb windows corresponding to 24 annotated genes (and a number of novel, uncharacterised genes) (S6 Table). Intersecting these windows with within-river candidate regions highlighted that most HP-LP associated candidates reflected within-river selection scan outliers in one to three rivers (S7 Table). Selection scans may overlook some of our association outlier windows because differentiation at these loci may be moderate, but rather we are detecting consistent allele frequency changes in the same direction between HP-LP comparisons. Many of the associated windows mapped to a previously unplaced scaffold in the genome (000094F), but we were able to place this at the start of chromosome 20 along with some local rearrangements (S10 Fig) using previously published HiC data [67]. From here on and in Fig 3C and 3D, we refer to this new arrangement for chromosome 20 and scaffold 000094F as chromosome 20.

The clusters on chromosomes 8 and 20 exhibited multiple 10kb windows above the 99.9% quantile of window-averaged BF scores (chr8 = seven windows, chr20 = 54 windows), suggesting larger regions associated with HP-LP adaptation in multiple rivers (Fig 3D). In particular, the region at the start of chromosome 20 spanned several megabases with two distinct peaks. The entire chr20 region also exhibited some evidence of selection in four of the five rivers (all but Madamas; S7 Table), whereas the chr8 region only had evidence of selection in Guanapo and Aripo. Further, the larger of the chr20 peaks reflected the strongest region of differentiation in the Aripo river (S11 Fig) (which was minimally differentiated genome-wide, S9 Fig). Based on the substantially stronger evidence of convergence at the chr20 region, compared with the second largest cluster of outliers on chr8, we explored the chr20 region further to evaluate: which rivers showed evidence of HP-LP differentiation within these regions; by what mode of convergence these regions had evolved under; and their gene content and probable candidates for HP-LP phenotypes. Gene content and selection scan overlap for the chr8 region, and other BayPass outliers, is available in S6 and S7 Tables.

## Candidate region on chromosome 20

Visualisation of genotypes (Fig 4A) illustrated extended haplotype structures that were consistent with haplotypes spanning the entire chromosome 20 candidate region (Fig 3D). Interestingly, two of the three Caroni LP populations (GLP and TACLP) were fixed or nearly fixed for homozygous ALT haplotypes across the region (Fig 4A). We will refer to this entire region (~0–2.5 Mb) as the 'CL haplotype' (Caroni LP haplotype). The other haplotype we will refer to

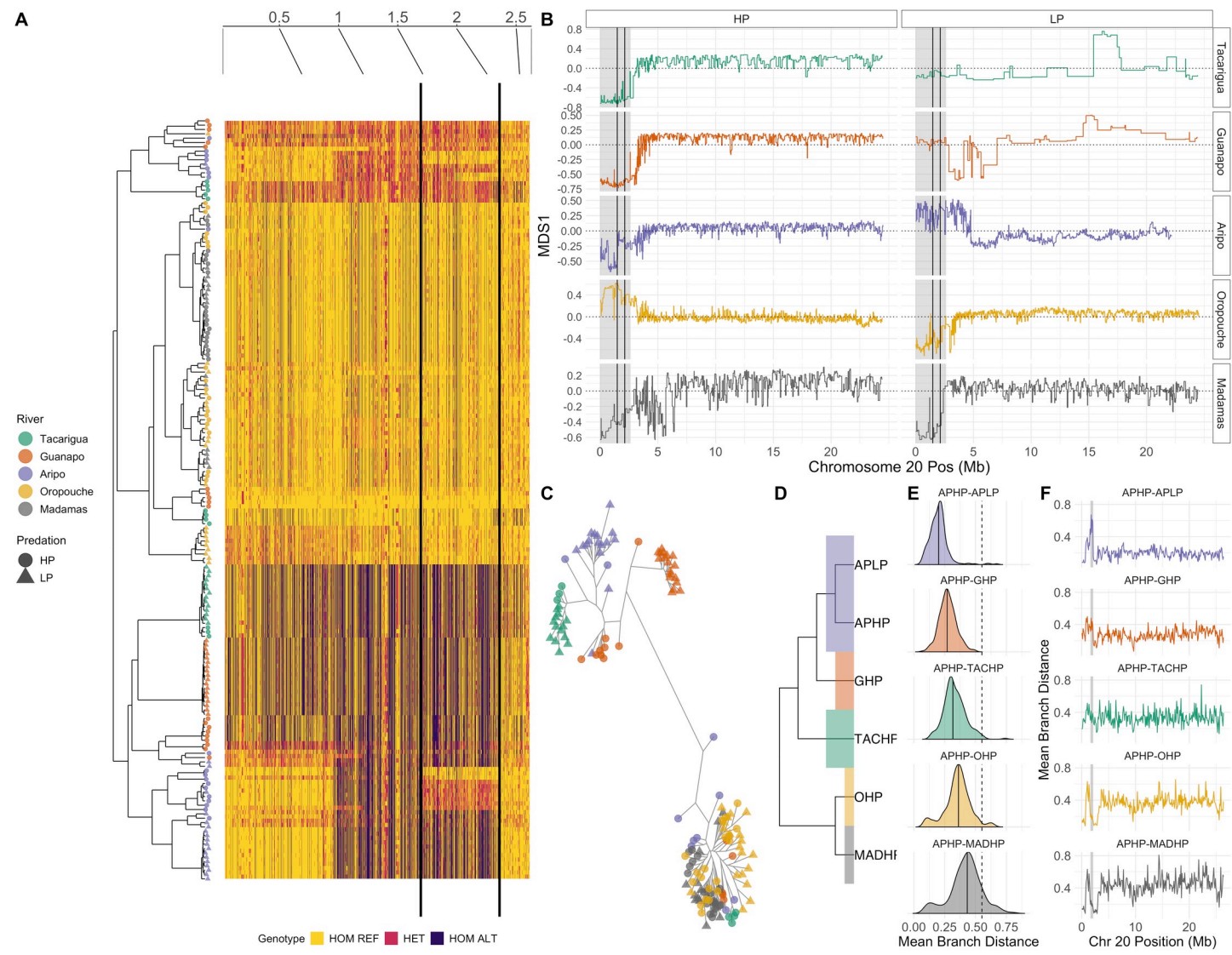

**Fig 4. Evidence of divergence along the CL haplotype (ALT alleles, chr20:1–2633448).** Genotypes for each individual plotted according to hierarchical clustering of PCA (A). Solid black lines denote the CL-AP region, corresponding to the strongest peak of HP-LP association in the dataset. (B) The CL haplotype region (shaded grey) shows evidence of segregated local ancestry in comparison to the rest of chromosome 20, according to MDS scores derived from local PCA. The CL-AP region is again shown between solid black lines. MDS1 = 0 is shown as a dotted line in each panel. A lack of signal on MDS1 for the CL region in Guanapo LP (GLP) and Tacarigua LP (TACLP) reflects that this region is fixed. (C) Unrooted maximum-likelihood tree of homozygous individuals (haplogroups) across the CL-AP region, highlighting a major phylogenetic branch separating Caroni LP individuals from both Caroni HP individuals (homozygous for the REF haplotype) and populations from outside the Caroni drainage. (D-F) Branch length analysis of CL-AP region relative to the whole of chromosome 20. Density distributions (E) show mean branch lengths between populations in 100kb windows across chromosome 20 (F), with the median highlighted in each. The mean branch length between the REF and CL haplotype at the CL-AP region in Aripo is marked on all density distributions as a dashed line. Density distributions are ordered vertically according to increasing phylogenetic distance, as summarised in the adjoining tree (D). The CL-AP region is highlighted in panel F as a grey rectangle.

as the 'REF haplotype' due to its closer similarity to the reference genome. Moreover, a subset of the candidate region was also nearly fixed in APLP (between black lines, between ~1.53–2.13 Mb, referred to as the CL-AP (CL Aripo) region Fig 4A). This CL-AP region corresponds to both the region of highest genome-wide divergence in Aripo (S11 Fig), and the largest peak in our HP-LP association analysis (Fig 3D).

We then assessed whether ancestry of our candidate region (Fig 4B) deviated from the rest of chromosome 20 using a local PCA. This approach is sensitive to inversions, changes in

recombination and gene density [68], which may explain why such a large region appears as HP-LP associated. This approach confirmed that the associated region exhibits distinct local ancestry in relation to the rest of the chromosome in all five rivers (Fig 4B), albeit with some idiosyncrasy. For example, MDS scaling along the major axis was broadly similar for GHP and TACHP (LP populations were fixed and therefore showed minimal signal), with the entire region segregated as a single block. Similar results were observed in Oropouche and Madamas. In Aripo however, smaller blocks within the region were the major drivers of local ancestry. Linkage analyses confirmed strong linkage across the several megabases spanning the HP-LP associated region in most of the sampled populations (S12 Fig).

These patterns may be consistent with a large inversion, polymorphic in Caroni HP populations but fixed in Caroni LP populations. Further structural variation (SV) or a recombination event between the CL haplotype and the REF haplotype may then have released the CL-AP region from the larger CL haplotype in Aripo only. We therefore explored the potential for inversions and SVs with our aligned read data using smoove [69] (v0.2.5) and Breakdancer [70] (v1.4.5). We did not find evidence for an inversion or an alternative SV spanning either the full CL haplotype (in Guanapo or Tacarigua) or around the diverged CL-AP region in Aripo. Interestingly however, this analysis of SVs did highlight that the strongest peak of HP-LP differentiation in the Oropouche river (chromosome 15 at approximately 5Mb, S13 Fig), was associated with a detected 1.1 kb deletion within the *B-cadherin* gene, and exhibited high HP-LP $F_{ST}$ (0.66). The lack of an inversion underlying the CL haplotype would agree with our local PCA results, where deviations were observed at the start of chromosome 20 in Madamas and Oropouche, despite the CL haplotype being absent.

To understand the mode of convergence at the CL-AP region, we reconstructed the phylogenetic history of the haplotype at this region. To start, we performed PCA over the CL-AP region, and found three clusters along a PC1 axis with large loading (PC1 = 57%, S14 Fig), consistent with individuals tending to either be homozygotes for either haplotype or heterozygotes. We used these clusters to define homozygous individuals and explored the phylogenetic history of these homozygote haplotypes following phasing. A maximum-likelihood tree using RAxML-NG (version 0.9.0) illustrated that the CL haplotype at the CL-AP region is phylogenetically distinct from the REF haplotype and separated by a long branch (Fig 4C). This clustering of CL haplotypes and REF haplotypes contradicts the neutral expectation that haplotypes should be predominantly structured within rivers. The clustering of Oropouche and Madamas individuals with Caroni REF haplotypes was surprising, and in stark contrast to the genome-wide structure that clearly separates rivers by drainage (Fig 1C).

To evaluate the relative age of these haplotypes, we compared the mean branch length between CL-REF homozygotes in the Aripo river at the CL-AP region to the distribution of mean branch lengths across chromosome 20 (Fig 4D, 4E and 4F) across the phylogeny. In Aripo (APHP-APLP), the CL-AP region is clearly more diverged than the rest of chromosome 20 (Fig 4E), and the mean branch length between the REF and CL haplotype was generally greater than the phylogenetic distance between APHP and all HP populations. This suggests that the CL and REF haplotype separation most likely predates the common ancestor of all five rivers. It is particularly interesting that given the CL haplotype may be reasonably old, it has been broken down into smaller regions only in the Aripo river, whereas Tacarigua and Guanapo maintain the full haplotype.

Within the CL haplotype region there are 56 annotated genes (S8 Table), several of which may have important roles in HP-LP phenotypes. Due to the elevated differentiation observed across the haplotype, it is difficult to pinpoint specific candidates. However, the breakdown of the haplotype in Aripo at the CL-AP region, corresponding to our association peak, provides a unique opportunity to narrow down candidate genes, given it is the only part of the larger

candidate region that is differentiated in all Caroni HP-LP comparisons. Interestingly, analyses of coverage across the CL-AP region uncovered repeatable low coverage in all Caroni LP populations that corresponded with deletions of several kb (viewed in *igv*) in the LP populations (S15 Fig). Five of these deletions overlapped with the *plppr5* gene, including a deletion subsuming the final exon of the gene (S15B Fig). This gene also spanned the HP-LP associated windows with the highest association scores (S6 Table). The adjacent *plppr4* gene included the individual SNP (000094F_0:556282) with the highest HP-LP association score of all SNPs in the genome (S15A Fig). This particular SNP was observed in the intron between the second and third exons of *plppr4*.

In summary, this region presents a fascinating example of genetic convergence of an ancestrally-inherited large haplotype among rivers in the Caroni drainage. As such, subsequent genetic divergence of HP-LP adaptation is observed in non-Caroni rivers due to the presumed loss of the CL haplotype in the lineage ancestral to non-Caroni rivers. Additionally, Aripo exhibits a unique signature of stronger HP-LP differentiation at a subregion of the haplotype due to a potentially more recent recombination event between the larger, diverged haplotypes themselves.

## Discussion

### Summary of results

Using a whole genome sequencing approach, we found a strong candidate haplotype for HP-LP convergence within the Caroni drainage of Northern Trinidad (the only drainage where we have multiple rivers sequenced). More generally, we found molecular convergence at specific loci is limited among rivers from different drainages. Further, we find evidence that convergence at the level of functional pathways among rivers may facilitate phenotypic convergence across all rivers. Our convergent LP candidate region exhibited a strong signal of divergent selection between HP-LP sites on chromosome 20. This region contains a large ancestral haplotype fixed or nearly fixed in LP populations in all three Caroni rivers examined, and contained promising candidate genes for LP phenotypes. Our analysis of population structure, admixture, and demographic histories across Northern Trinidad suggest that the reduced re-use of the same alleles among drainages may be due to strict structuring of genetic variation between some rivers and recurrent bottlenecks during the founding of LP populations from HP sources. Combined, these processes limit shared ancestral genetic variation from which convergent genetic adaptation may occur. This is not true for all rivers however, with strong evidence of gene flow taking place between rivers outside of the Caroni drainage.

### Convergence at the CL haplotype

Our analyses highlighted the 'CL' haplotype on chromosome 20 as a clear outlier in terms of association between allele frequencies and HP-LP classification. Within the CL haplotype, the 'CL-AP' region represented the strongest candidate for convergent HP-LP adaptation due to its particularly strong HP-LP association peak and high within-river differentiation in Aripo.

Recent empirical work has demonstrated the importance of large haplotype regions containing many genes in convergent evolution. In *Littorina*, large divergent haplotypes are maintained by inversions in crab vs wave ecotypes [71,72]. Similarly, sunflower species repeatedly experience selection on large haplotypes [23], most, but not all, of which involve inversions. This recent empirical evidence suggests a fundamental role of large haplotype blocks in adaptation by bringing together and maintaining clusters of adaptive alleles, although we cannot rule out genetic draft occurring around a single functional locus within the CL haplotype. We did not detect evidence of inversions within this region, but given that we detect deviations in

local ancestry in all rivers (Fig 4B) relative to the rest of the chromosome, and the acrocentric nature of guppy chromosomes [73], it is possible that recombination is reduced over the CL haplotype due to proximity to the telomere. This mechanism could maintain this haplotype in the absence of an inversion. We noted however that, whilst the CL haplotype was fixed (at the CL-AP region) in Caroni LP populations, it was polymorphic in all Caroni HP populations. Large haplotypes may bring together beneficial alleles but they can generate constraint if different loci within the haplotype experience contrasting selection. Breakdown of the haplotype, potentially involving double crossover events in Aripo, may have reduced constraints associated with genetic background. Subsequently, this may be why we observe stronger HP-LP differentiation at the CL-AP region uniquely in this river.

Within the CL-AP region we highlighted the *plppr5* and *plppr4* genes as strong candidates for HP-LP adaptation. These genes correspond to the strongest signals of HP-LP association within the CL-AP region, and in particular the *plppr5* allele on the CL haplotype is associated with an exon-subsuming deletion (S15B Fig). There is limited functional evidence for these genes, but evidence suggests a possible role in growth and body size. Transcriptome analysis has shown that *PLPPR4* is among genes upregulated in slow-growth vs fast-growth Jinghai Yellow Chicken chicks [74], and transgenic mice studies have demonstrated phenotypic effects of *Plppr4* on body size and growth phenotypes [75]. In humans, *PLPPR4* expression is limited to the brain, but *PLPPR5* expression occurs more broadly.

Across the CL haplotype region, HP populations were polymorphic, which is likely why we fail to detect this region as within-river outliers in Guanapo and Tacarigua. Further, these rivers have small LP populations and elevated signatures of genome-wide drift. That the CL haplotype region is variable within HP populations suggests that there is selection on the CL haplotype in LP populations but not against it in HP populations, or that downstream-biased migration is strong enough to maintain the CL haplotype in downstream HP populations. At the CL-AP region, we note that haplotypes derived from Oropouche and Madamas cluster with REF haplotypes from Caroni (Fig 4C), despite genome-wide data suggesting the split between Caroni and Northern/Oropouche drainage rivers is the deepest in our data (Fig 1C and 1D; although not in Fig 1B). Such patterns can arise when diverged haplotypes are introgressed from more ancient lineages, or even different species, as observed in flatfish [76], sunflowers [23], and *Heliconius* butterflies [24]. We found, however, that branch lengths between the CL and REF haplotype clusters were in keeping with branch lengths within the phylogeny (Fig 3D, 3E and 3F), suggesting it is unlikely that the CL haplotype evolved elsewhere in an unknown lineage before more recently being introgressed into the Caroni drainage. This pattern suggests the CL haplotype may have evolved prior to the splitting of Caroni, Northern, and Oropouche drainage lineages, but is subsequently absent due to loss through drift or selection outside of the Caroni drainage. Alternatively, the CL and REF haplotypes could represent the ancestral Caroni and Northern/Oropouche drainage haplotypes respectively, and the current distribution may reflect introgression of the REF haplotype into the Caroni drainage and subsequent gene flow among HP populations within the Caroni drainage. This would however not explain the strong differentiation between APHP and APLP at the CL-AP region, and seems unlikely based on limited HP-HP gene flow in the Caroni river.

## Population structure and limitations on the sharing of adaptive variation

By using whole-genome data, we were able to explore in fine-detail how genetic variation is structured and distributed across natural guppy populations across Northern Trinidad. Our observations support previous work suggesting downstream-biased migration, strong drainage-based structuring, and variable gene flow among rivers [77,78]. Using within-river

demographic analyses we also found downstream-biased migration in all rivers, but with variable rates, and three LP populations experiencing bottlenecks; these bottlenecks likely represent historical founding bottlenecks as opposed to recent crashes [77]. Such demographic processes, if strong enough, can obscure signals of genetic convergence, or even produce false-positives by manipulating the relative efficacy of selection and neutral processes across the genome [79].

A particularly interesting question within this system is whether LP populations are derived from HP sources, or vice versa. Whilst neither contemporary HP and LP populations represent the shared ancestral population, the extent to which HP and LP populations have drifted from the common ancestral state is highly suggestive of which was most similar to the ancestral state. Our data thus strongly support the direction of LP evolution from HP (or HP-like) sources, as LP populations are observed at more extreme regions of PCA space (Fig 1D); indicative of excessive drift. HP-LP population splits were estimated as several thousands of years old, however caution should be taken when interpreting variation among rivers as greater within-river migration will downwardly bias divergence estimates. However, the range observed among rivers (1,999–29,922 years) provides a useful ballpark figure for the ages of the HP-LP systems generally, and suggests HP-LP populations may be older than the estimate by Endler [80] of less than 1000 years. The time interval estimated for splits within all rivers here overlaps with the end of the last glacial maximum, when Trinidad experienced substantial change due to flooding and separation from the South American mainland. In addition, our estimates of cross-drainage divergence are considerably younger than the 0.6–1.2 mya estimates presented by mitochondrial sequences [60], and provide little evidence for the suggestion that Oropouche drainage populations are a separate species [81]. Estimates of divergence times based on mitochondrial and nuclear markers can differ for many reasons [82], including non-neutrality of the mitochondrial genome, different effective population sizes of the mitochondrial and nuclear genome, or sex-biased migration. Generally, as our divergence times are based on SNPs located genome-wide as opposed to a single locus, our estimates may be considered more robust.

In our introgression analyses, we observed evidence of introgression in the Aripo HP population from the Oropouche and Madamas rivers. Aripo represents the most easterly river within the Caroni drainage, whilst Oropouche/Quare is the most westerly river in the Oropouche drainage (Fig 1A). Thus, admixture between these populations may be possible, and indeed has been suggested elsewhere [54,78,83]; likely facilitated by flooding during the wet season. The Aripo river may be particularly susceptible to contemporary human translocations of guppies from across Trinidad, as it is heavily involved in active research including experimental introductions. We also found strong introgression between OLP and (most likely) MADLP. Introgression between upstream LP populations has been reported between the Paria and the Marianne rivers in the Northern drainage [84], but is surprising here given these rivers are in separate drainages. Despite this however, genetic convergence was not more pervasive between Oropouche and Madamas than other river comparisons (Fig 2A), suggesting that other contingencies such as contrasting selection among rivers or genetic redundancy are probably important in this system.

Contrasting demographic contexts can influence the genetic architecture of traits or the regions of the genome where adaptive alleles reside, and may be important here given variable connectivity of HP-LP pairs. Theory predicts that with increasing sympatry, if multiple genetic routes to a phenotype exist then selection should favour simpler genetic architectures (e.g few loci of larger effect) because they are less likely to be broken down by introgression than complex genetic architectures (e.g. many loci with small effect) [85,86]. Empirical support from cichlid species pairs suggests this is the case for male nuptial colour traits in sympatric vs

allopatric pairs [87]. Given demographic histories vary between our HP-LP populations (Table 1), these natural conditions may moderate the selective benefits of different genetic routes to phenotypes.

Our population structure results have important implications for the likelihood of observing genetic convergence because they identify constraints on the sharing of ancestral genetic variation through LP founding bottlenecks and limitations on adaptive variants being shared among rivers [3]. These constraints are particularly obvious with the limited sharing of genetic variation among LP population in their 2dsfs (S4 Fig). The exception here is OLP-MADLP, where introgression, most likely into OLP, has led to limited private variation within MADLP. Such structuring of genetic variation, typical of riverine populations, stands in contrast to other prominent systems with abundant evidence of genomic convergence, such as sticklebacks [88] and atlantic herring [89], where largely panmictic marine populations share much adaptive variation. Standing genetic variation is a major contributor of adaptive variation [90–92], and the sharing of this variation among lineages acts a significant contingency to genetic convergence [93] in varied systems including fish [25,94] and insects [95].

### Functional convergence of the cadherin-signaling pathway

Redundancy in the mapping of phenotype to genotype is expected for highly polygenic traits, whereby many loci may be adapted to modify a phenotype, and in instances where phenotypes are derived from complex functional pathways [28]. Indeed, convergence at the level of pathways has been described for human pygmy phenotypes [15] and hymenopteran caste systems [14]. In our selection scans, we found a greater proportion of genes associated with cadherin-signaling than expected across five replicated datasets, suggesting this pathway may be under selection at different genes in all rivers. Cadherin genes *cadherin-1* and *B-cadherin* have previously been detected as under selection in experimentally transplanted LP populations derived from GHP [56], however these specific genes, whilst also detected here (S13 Fig), were omitted from our pathway analysis due to many-to-many orthology with zebrafish genes. Genes in this pathway have important roles in cell-cell adhesion, and are associated with tissue morphogenesis and homeostasis by mediating interfacial tension and orchestrating the mechanical coupling of contact cells [96]. Cadherin signaling pathways interact with the signaling of various growth hormones and are involved in differential growth phenotypes. For example, cadherin signaling genes were differentially expressed between transgenic and wild-type coho salmon with divergent muscle fibre phenotypes that affect growth and the energetic costs of maintenance [97]. Cadherin genes are also expressed during oogenesis in *Drosophila* [98]. Assessing the functional roles of the cadherin signaling genes identified in our study (S5 Table) is beyond the scope of this work, but in identifying this pathway across rivers we provide evidence for a potentially shared mechanism by which HP-LP phenotypes may similarly evolve across Northern Trinidad.

### Concluding remarks

We have investigated whether convergent HP-LP phenotypes that have evolved repeatedly within rivers across Northern Trinidad are underpinned by convergent genetic changes. We found convergence of genetic pathways, not specific genes across drainages. This is in keeping with recent work suggesting a predominant role of shared standing genetic variation in driving convergent changes at the gene-level, a mechanism that is restricted in natural guppy populations by limited between-river gene flow and recurrent founding bottlenecks during LP colonisations. Within our drainage with multiple rivers sampled, we did however find convergent evolution of a large haplotype region nearly fixed in Caroni LP populations that is likely

derived from shared ancestral variation between these populations. Additional sampling of multiple rivers within the same drainage could identify comparable drainage-specific candidates in the Northern and Oropouche drainages. These results provide a comprehensive, whole-genome perspective of genetic convergence in the Trinidadian guppy, a model for phenotypic convergence.

## Methods

### Sampling, sequencing and SNP calling

Individuals were sampled from naturally-occurring downstream HP and upstream LP environments in rivers from each of Northern Trinidad's three drainages (Fig 1A and S1 Table) between 2013 and 2017. Three of these rivers (Aripo, Guanapo, Tacarigua) share a drainage (Caroni), whilst Oropouche (Oropouche) and Madamas (Northern) are found in separate drainages. Numbers of individuals from each population were: TACHP = 12, TACLP = 14, GHP = 19, GLP = 18, APHP = 19, APLP = 19, OHP = 19, OLP = 20, MADHP = 20, MADLP = 16. Samples were stored in 95% ethanol or RNeasy at 4° C prior to DNA extraction. Total genomic DNA was extracted using the Qiagen DNeasy Blood and Tissue kit (QIAGEN; Heidelburg, Germany), following the manufacturer's guidelines. DNA concentrations $\geq$ 35ng/μl were normalised to 500ng in 50μl and were prepared as Low Input Transposase Enabled (LITE) DNA libraries at The Earlham Institute, Norwich UK. LITE libraries were sequenced on an Illumina HiSeq4000 with a 150bp paired-end metric and a target insert size of 300bp, and were pooled across several lanes so as to avoid technical bias with a sequencing coverage target of $\geq$10x per sample. Data from the Guanapo and Oropouche rivers has been previously published as part of Fraser et al. [67].

Paired-end reads were quality-controlled with fastQC (v0.11.7) and trimmed with trim_galore (v0.4.5) before being aligned to the long-read, male guppy genome assembly [67] with bwa mem (v0.7.17). Appropriate read groups were added followed by alignment indexing, deduplication and merging to produce final bams. Merged, deduplicated alignments were recalibrated using a truth-set of variants generated from high-coverage, PCR-free sequencing data from 12 individuals [67]. GVCFs were produced using GATK's (v4.0.5.1) HaplotypeCaller and consolidated to chromosome/scaffold intervals with GenomicsDBImport prior to genotyping with GenotypeGVCFs.

SNPs were filtered on the basis of QD < 2.0, FS > 60.0, MQ < 40.0, HaplotypeScore > 13.0 and MappingQualityRankSum < -12.5 according to GATK best practices. We retained only biallelic sites with a depth $\geq$ 5. SNPs were also removed if missing in > 50% of individuals within a population, if they were not present in all ten populations, and had a minor allele frequency < 0.05 (relative to all individuals). This produced a final dataset of 3,033,083 high-quality SNPs.

### Population structure and introgression

Principal Component Analysis (PCA) was performed over all ten populations using a linkage-pruned (—indep-pairwise 50 5 0.2) set of SNPs (N = 217,954) using plink (v2.00).

Prior to further estimates of population structure, chromosomes and scaffolds were phased individually with beagle (v5.0; [99]), which performs imputation and phasing, and then phased again using shapeit (v2.r904; [100]) making use of phase-informative reads (PIR). This combined method has been effective elsewhere [101]. Phased chromosomes were re-merged into a single file for analysis with fineSTRUCTURE (v4.0.1; [58]). FineSTRUCTURE first makes use of chromosome painting before assessing admixture based on recombinant haplotype sharing

among all individuals. FineSTRUCTURE was run with a uniform recombination map and an inferred *c*-value of 0.344942 with successful convergence.

For analyses of divergence times and introgression, we used previously published [102] sequencing data from six *P. wingei* individuals (3 males, 3 females). We aligned these to the male guppy genome using the protocol above. SNPs were called in a new cohort alongside guppy individuals and filtered as above, with the exception that a 1% MAF filter was applied. This produced a second VCF with a total of 4,829,912 SNPs.

For divergence estimates, we used SNAPP [57] and applied the workflow outlined by Stange et al. [103] using scripts available at github.com/mmatschiner/tutorials/tree/master/ divergence_time_estimation_with_snp_data. This is an implementation of SNAPP that includes a time-clock model that can be calibrated based on information within the phylogeny. To calibrate our phylogeny, we rooted the divergence between *P. reticulata* and *P. wingei* at 3.41 mya and a standard deviation of 0.329 based on estimates from five studies collated by Time-Tree [59]. TimeTree confidence intervals are normally distributed, so we used a normal distribution with mean 3.41 and sd 0.329. For input, we included SNP data from 36 individuals: six from the outgroup and three from each of the ten populations with the lowest % missing data. We filtered for SNPs with no missing data among these 36 individuals and thinned SNPs to one every 50kb with vcftools—thin 50000. Of these SNPs, we selected the top 1000 SNPs that were informative for populations based on $F_{ST}$. This step was necessary in order to achieve convergence in trees, given these populations are relatively young and there is a greater expectation for gene flow compared with species-level phylogenetic inference. We performed three runs with different starting seeds, each with 1,000,000 MCMC iterations. Convergence was achieved after burn-ins of approximately 500,000, 200,000, and 100,000 iterations for the three runs respectively based on Tracer (v1.7.1) [104] summaries. We removed these burn-ins and merged the converged 500,000, 800,000 and 900,000 MCMC iterations for a total of 2,200,000 MCMC iterations (sampled every 1000 iterations), which yielded ESS for all parameters >320. The maximum clade credibility (MCC) tree (Fig 1B) and statistics were calculated with FigTree (v1.4.4). Node ages were taken as the mean height of nodes with 95% HPD intervals.

To assess introgression, we calculated D-statistics, f4-ratios and Fbranch statistics with Dsuite [61]. We highlighted introgression candidates as trios with significant D values (Bon-ferroni-corrected *p*-value < 0.001) and f4-ratios > 0.05 between populations from different rivers i.e. we retained trios for which P1 and P2 were assigned to the same river by including the tree structure to examine the possibility of introgression between rivers.

## Demographic inference

We explicitly modelled the demographic history of each HP-LP population pair using *fastsim-coal2* [63]. Fastsimcoal2 uses a continuous-time sequential Markovian coalescent approximation to estimate demographic parameters from the site frequency spectrum (SFS). As the SFS is sensitive to missing data, a —max-missing filter of 80% was applied to each population VCF containing both monomorphic and polymorphic variants, and to remove LD, the VCF was thinned at an interval of 20kb using *vcftools* [105]. For each HP-LP population pair, we generated a folded two-dimensional frequency spectrum (2dSFS) using the minor allele frequency, which were generated via projections that maximised the number of segregating sites using easySFS (https://github.com/isaacovercast/easySFS).

Five demographic models were used to explore the demographic history of the population pairs, all of which contained a uniform distribution divergence of 1 to 6e7 and log uniform distribution $N_e$ of 1 to 50000: Model A) HP-LP split with no population growth; Model B) HP-LP split with no population growth and a post-founding bottleneck in the LP population; Model

C) HP-LP split with historical population growth in HP; Model D) HP-LP split with historical population growth in HP and a post-founding bottleneck in the LP population; Model E) HP-LP split with bottlenecks in both LP and HP populations. All five models were also run with the inclusion of a migration matrix between LP and HP with a log uniform distribution of 1e-8 to 1e-2. Fastsimcoal2 was used to estimate the expected joint SFS generated from 100 independent runs, each consisting of 200,000 simulations per estimate (-n), generated by 100 ECM cycles (-L). Model choice was assessed by computing the log likelihood ratio distributions based on simulating 100 expected SFS from the run with the lowest delta (smallest difference between MaxEstLhood and MaxObsLhood) as per Bagley et al. [106]. The most likely model was selected for each population pair and the run with the lowest delta likelihood was used as input for bootstrapping by simulating 100 SFS. We report the median and 95% confidence intervals for $N_e$ and probabilities of migration as provided by bootstrapping.

## Scans for selection

We used three approaches to scan the genome for signatures of selection between HP-LP populations. We first estimated XtX (a Bayesian approximation of $F_{ST}$) within each river using BayPass [64], which has the advantage of including a genetic covariance matrix to account for some demographic variation. Genetic covariance matrices were estimated using LD-pruned (plink—indep-pairwise 50 5 0.2) VCFs for each river, and averaged over 10 independent runs. We determined a significance threshold within each river by simulating neutral XtX of a POD sample of 10,000 SNPs with the *simulate.baypass()* function in R. We then marked 10kb windows as outliers if their mean XtX value exceeded the 0.95 quantile of the neutrally simulated distribution. Secondly, absolute allele frequency differences (AFD) were estimated by taking allele counts from each population (vcftools —counts2) and estimating frequency changes per SNP. To convert per SNP values to 10kb non-overlapping windows we removed invariants within each river, calculated the median AFD, and filtered windows that contained fewer than 6 SNPs. We marked outliers as windows above the upper 0.95 quantile, or with an AFD > 0.5 if this quantile was > 0.5. The linear association with AFD and differentiation, compared to the non-linear equivalent for $F_{ST}$ makes this comparable measure of allele frequency change more interpretable [65]. A cutoff of AFD = 0.5 therefore represents the minimum by which to observe a change in the major allele between HP and LP. Finally, we estimated the extended haplotype homozygosity score XP-EHH between each river with selscan (v1.2.0a; [107]) and normalised in windows of 10kb. We limited this analysis to chromosomes and scaffolds > 1 Mb in size, due to extreme estimates on smaller scaffolds distorting normalisation. Outliers were marked as those with normalised XP-EHH > 2.

Enrichment analyses were performed by extracting one-to-one zebrafish orthologs for guppy genes in outlier windows using Ensembl's BioMart (release 101; [108]). Orthologs were then assessed for enrichment within rivers by comparing outlier genes against a background set of all genome-wide guppy-zebrafish one-to-one orthologs using PantherDB [109]. Results from all rivers were then compared in a single analysis to assess the above-expected enrichment of functional groups across the entire dataset. We performed random permutations (N = 10,000) to draw equivalently sized within-river sets of outliers with weightings based on the number of genes within each functional pathway group within the guppy-zebrafish ortholog background gene set. Based on permuted random outlier sets, we then assessed the by-chance likelihood of observing within-river enrichment >1 for all five or any four rivers for each of the functional groups where this had been observed (groups plotted in Fig 3B).

For associated allele frequency changes with HP-LP classification, we also applied BayPass's auxillary covariate model, which associates allele frequencies with an environmental covariate

whilst accounting for spatial dependency among SNPs with an Ising prior (-isingbeta 1.0). We used a genetic covariance matrix including all 10 populations estimated as above. We split our SNP data into 16 subsets, allocating alternate SNPs to each subset such that all subsets included SNPs from all regions of the genome, and merged outputs as recommended in the manual. We averaged per SNP BayesFactor scores within 10kb and marked as outliers those above the 0.999 quantile. We also explored alternative windowing strategies here, such as marking outlier SNPs above the 0.999 quantile and determining window significance as windows with significantly more outlier SNPs than a binomial (99.9%) expectation, however this made a minimal impact on which windows were deemed outliers.

Reduced SNP counts within 10kb windows can increase variance and lead to spurious outlier detection. To examine whether our focal windows were affected by this, the median number of SNPs in windows that were called as overlapping selection scan outliers or BayPass outliers was compared against a permuted distribution (10,000 permutations). Permutations randomly sampled N 10kb windows from the whole genome, where N = the number of overlapping selection scan outlier windows or BayPass outlier windows respectively. The median SNP count in each set of outlier windows was compared against the permuted null distribution of median SNP counts.

### Characterisation of structural variants

To assess whether candidate regions may contain structural variants, particularly inversions, we first used local PCA within each population for each chromosome with the R package *lostruct* (v0.0.0.9) [68]. This method explores phylogenetic relationships between individuals using windows of N SNPs along a chromosome, and then uses multidimensional scaling (MDS) to visualise chromosomal regions that deviate from the chromosomal consensus among individuals. We filtered each population's chromosome for invariants, prior to running local PCA in windows of 100 SNPs. For each run, we retained the first two eigenvectors (k = 2) and computed over the first two PCs (npc = 2).

We used two methods to call structural variants from our final bam files: smoove (v0.2.5; [69]) (a framework utilising lumpy [110]) and Breakdancer (v1.4.5; [70]). For both methods we called structural variants across all HP and LP individuals within a river. For smoove variants, we excluded repetitive regions of the genome prior to variant calling, and filtered the subsequent VCF for SVs marked as imprecise; <1kb in size; less read pair support (SU) than the per-river median; without both paired-end and split-read support (PE | SR = = 0). Breakdancer was run per river, per chromosome, with results filtered for SVs <1kb in size; less support than per river, per chromosome median support; quality < 99. We calculated $F_{ST}$ using smoove VCFs within each river to explore structural variants that may have diverged within rivers. To validate SVs of interest, we plotted all bam files within a river using samplot [111] and visualised regions in *igv*.

### Phylogenetic relationships of haplotypes

To examine phylogenetic relationships of the CL haplotype, we calculated maximum likelihood trees among homozygote haplogroups using RAxML-NG (v0.9.0; [112]). We limited this analysis to the CL-AP region, which was subsumed within the larger CL haplotype in Guanapo and Tacarigua and included the strongest region of HP-LP association (CL-AP region). Haplogroups were defined on the basis of PC1 clusters (S14 Fig). We retained a random haplotype from homozygote individuals from all populations and constructed a tree using the GTR + Gamma model with bootstrap support added (500 trees) with a cut-off of 3%.

To estimate the relative age of the CL haplotype, we calculated the branch distance at the CL-AP region between APHP and APLP, given these populations are strongly differentiated

here. We then compared this APHP-APLP CL-AP branch distance to the distribution of branch distances calculated along the entire chromosome 20. Branch distances were taken as the mean distance between individuals from either population, and were calculated in 100kb windows. Within each window, a bioNJ tree was produced with the *poppr* [113] function *aboot()* and branch distance was estimated as Nei's distance (nei.dist). We repeated this process at increasing phylogenetic distances by comparing APHP with GHP, TACHP, OHP and MADHP.

## Supporting information

**S1 Fig. Bootstrapped density tree of all sampled MCMC iterations for the rooted phylogeny of the ten guppy populations and the outgroup *P. wingei*.** Grid lines denote divergence times in millions of years (mya). Black trees are those that share the most common topology. Orange trees were the second most common topology, and blue the third. The consensus tree is marked as a solid green tree. Trees plotted are based on merged post-burn-in trees sampled every 1000 iterations from a total of 2,200,000 MCMC iterations.
(PDF)

**S2 Fig. Repeated PCA, each of which involved the removal of individuals from a Caroni river.** These results demonstrate that our population structure results are unlikely to be driven a sample size bias towards Caroni rivers. Each panel shows PC1 and PC2 with % variance explained. Point colour and shape reflect river and predation environment respectively. Ellipses show 95% confidence around river groups. In all cases, PC1 reflects the split between Caroni and non-Caroni populations, and PC2 shows structure within the Caroni drainage.
(PDF)

**S3 Fig. Map of sampling rivers in Northern Trinidad alongside topography.** Sampling rivers are coloured according to legend and other major rivers are coloured light blue. Topography ranges from low altitude (dark fill) to high altitude (light fill). Upstream regions of rivers in the western Caroni drainage (Tacarigua, Guanapo and Aripo) are flanked by mountain ranges, likely preventing gene flow occurring between these rivers. The coastline shapefile was sourced from OpenStreetMap (openstreetmap.org; CC BY-SA 2.0), rivers were added manually and are available as shapefiles at Zenodo doi: 10.5281/zenodo.4740381, and elevation rasters were sourced from the US Geological Survey (earthexplorer.usgs.gov; SRTM; US Public Domain).
(PDF)

**S4 Fig. Two-dimensional site frequency spectra (2d-sfs) between all pairwise LP-LP comparisons.** This figure demonstrates excesses of private variation as densities limited to the first column or first row. Beyond these regions of the sfs, genetic variation is typically limited to regions close to the first row or column, representing sites that are at low frequency in either population. These signatures are emblematic of minimal shared genetic variation among these populations. The exception to this pattern is MADLP-OLP, where the excess of OLP private alleles in the first column suggests gene flow in direction of MADLP > OLP. In each sfs, the frequency of sites in each population is illustrated from 0 to 2 N, where N is the number of individuals in each population. Projections have been scaled so each LP population has the same value of N (which distorts the shape of SFS across rows/columns. Each cell within these 2dsfs therefore shows the density (log-transformed) of SNPs with relevant allele counts in each population. Cells within the first column and first row show private alleles that are absent in one population (allele count of 0). Grey cells are missing data, where no SNPs are found at allele counts of x and y in LP and HP populations respectively.
(PDF)

**S5 Fig.** Distributions of selection scanning methods within each river and their associated outlier cut-offs for AFD (A), XP-EHH (B) and XtX (C).
(PDF)

**S6 Fig. Comparison of SNP counts in outlier windows.** SNP counts within all overlapping selection scan outlier windows (A) (within-river outlier for two or more selection scans in >1 river) and BayPass outlier windows (B) (>99.9% quantile). The first row shows the distribution of observed SNP counts within outlier windows. The second row shows the permuted distribution of median SNP counts for 10,000 randomly drawn window sets from the total genome set, each of which contains N windows, where N = the number of observed outlier windows in the first row. The median SNP count of the observed outlier windows is shown in each panel as a solid vertical line. Under a one-tailed hypothesis, respective permuted p-values were 0.0172 (A) and 0.0176 (B).
(PDF)

**S7 Fig. Genome-wide AFD results for 10kb windows. Panels represent the 23 chromosomes in the guppy genome.** Each row represents the change in the absolute allele frequency for 10kb windows between HP and LP populations in a different river. Chr20 has been updated to include the unplaced scaffold 000094F. The horizontal line in each row denotes river-specific outlier cutoffs.
(TIF)

**S8 Fig. Genome-wide XP-EHH results for 10kb windows. Panels represent the 23 chromosomes in the guppy genome.** Each row represents the normalised score for XP-EHH, which compares extended haplotype homozygosity between HP and LP populations within rivers (absolute-transformed). Chr20 has been updated to include the unplaced scaffold 000094F. The horizontal line in each row denotes the outlier cutoff = 2, analogous to a Z-score > 2 reflecting approximately p = 0.05 following normalisation.
(TIF)

**S9 Fig. Genome-wide XtX results for 10kb windows. Panels represent the 23 chromosomes in the guppy genome.** Each row represents the XtX score (a Bayesian analogue of FST, describing relative genetic differentiation) for 10kb windows between HP and LP populations in a different river. Chr20 has been updated to include the unplaced scaffold 000094F. The horizontal line in each row denotes river-specific outlier cutoffs, calculated according to neutral simulations of XtX within each river.
(TIF)

**S10 Fig. HiC contact information used to place scaffold 000094F_0.** Chromosome 20 co-ordinates are displayed along the upper axis, and scaffold 000094F_0 co-ordinates along the vertical axis. Scaffold 000094F_0 (total length = 1797025 bp) was reversed and inserted at position 836423 on chromosome 20. The proceeding scaffold on chromosome 20 (836423–3164071) was also inverted to reflect higher contact between the start of 000094F_0 and the chr20 region around 3164071.
(PDF)

**S11 Fig. XtX results for 10kb windows along chromosome 20 for each river.** Each row represents the XtX score (a Bayesian analogue of FST, describing relative genetic differentiation) for 10kb windows between HP and LP populations in a different river. Chr20 has been updated to include the unplaced scaffold 000094F. This figure highlights the location of a peak

of strong HP-LP differentiation int the Aripo river.
(PDF)

**S12 Fig. Linkage plots across chromosome 20 (following merging of chromosome 20 and scaffold 000094F_0).** Dark regions highlight elevated linkage ($R^2$), and light regions are low linkage. These plots highlight elevated linkage disequilibrium among SNPs at the start of chromosome 20 in several rivers.
(TIF)

**S13 Fig. Structural variant (SV) $F_{ST}$ and SNP XtX along chromosome 15 in the Oropouche river.** SVs were called between HP and LP populations, highlighting concordance between SV and SNP peaks at ~5 Mb (A). These peaks corresponded with the *B-cadherin* gene (B) in this region. The SV points at this peak correspond with the breakpoints of a 1,097 bp deletion detected using the software smoove.
(PDF)

**S14 Fig. PCA analysis across the CL-AP region for all individuals.** PCA highlights three clusters corresponding to homozygotes (REF and CL), and heterozygotes. Dashed lines denote cut-offs used to define haplogroups. Point colour represents river, and shape represents predation.
(PDF)

**S15 Fig. Association between CL-AP region and *plppr4* and *plppr5*.** Summary of CL-AP region (A) and *plppr5* gene region (B) according to differences in coverage between HP/LP populations and HP/LP association scores per SNP (BF). Of particular note are a peak in HP/LP coverage ratio in Tacarigua, Guanapo and Aripo at ~1.9 Mb (overlapping the *plppr5* gene), and the SNP with the highest genome-wide HP/LP association score at ~2.06 Mb (overlapping the *plppr4* gene). The peak in HP/LP coverage overlapped with the last exon of *plppr5*, was driven by reduced coverage in LP populations, and was thus confirmed as a ~1kb deletion in the CL haplotype in all Caroni LP populations by visualising bams in igv.
(PDF)

**S1 Table. Sampling locations and evidence of HP-LP phenotypes from previous studies.**
(XLSX)

**S2 Table. Divergence time estimates between HP-LP populations within rivers, and among rivers from SNAPP.**
(XLSX)

**S3 Table. Dsuite results for significant introgression from within-river trios (trios where P1 and P2 are within-river HP-LP pairs).**
(XLSX)

**S4 Table. Counts and overlaps of outlier 10kb windows detected by each selection scan method within each river.**
(XLSX)

**S5 Table. Genes identified with roles in the cadherin-signaling pathway (based on one-to-one zebrafish orthologs).**
(XLSX)

**S6 Table. Genes associated with outlier windows based on HP-LP association analysis (>99.9% quantile).** Annotated genes without a common name are referred to as "Novel".
(XLSX)

**S7 Table. Intersection between BayPass HP/LP association outliers and outliers from selection scans.** For each selection scan (final candidate = evidence from >1 scan), each entry describes the rivers in which that genome window was highlighted as an outlier.
(XLSX)

**S8 Table. Annotated genes in the CL region.** Scaffold 000094F_0 is reversed to reflect its placement in chr20. The CL-AP region is highlighted in bold.
(XLSX)

## Acknowledgments

We thank the FIBR guppy team for assisting with field logistics. Joan Ferrer Obiol provided advice and support regarding phylogenetic and introgression analyses. Kim Hughes provided comments and thoughts on an earlier draft of the manuscript. HPC infrastructure support was provided by The University of Exeter's High-Performance Computing (HPC) facility (ISCA). DNA sequencing was performed by University of Exeter Sequencing Service (ESS). Batch submission scripts for *fastsimcoal2* analyses were kindly provided by Vitor Sousa.

## Author Contributions

**Conceptualization:** Bonnie A. Fraser.

**Data curation:** James R. Whiting, Josephine R. Paris, Mijke J. van der Zee, Paul J. Parsons, Bonnie A. Fraser.

**Formal analysis:** James R. Whiting, Josephine R. Paris, Mijke J. van der Zee.

**Funding acquisition:** Detlef Weigel, Bonnie A. Fraser.

**Investigation:** James R. Whiting, Josephine R. Paris, Bonnie A. Fraser.

**Methodology:** James R. Whiting.

**Project administration:** Bonnie A. Fraser.

**Supervision:** Bonnie A. Fraser.

**Validation:** James R. Whiting.

**Visualization:** James R. Whiting.

**Writing – original draft:** James R. Whiting, Josephine R. Paris, Bonnie A. Fraser.

**Writing – review & editing:** James R. Whiting, Detlef Weigel, Bonnie A. Fraser.

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
