## [Decision Letter · Decision Letter 0]

11 Jan 2021

Dear Dr Whiting,

Thank you very much for submitting your Research Article entitled 'Drainage-structuring of ancestral variation and a common functional pathway shape limited genomic convergence in natural high- and low-predation guppies' to PLOS Genetics.

The manuscript was fully evaluated at the editorial level and by three independent peer reviewers. The reviewers appreciated the attention to an important problem, but raised some substantial concerns about the current manuscript. Based on the reviews, we will not be able to accept this version of the manuscript, but we would be willing to review a much-revised version. We cannot, of course, promise publication at that time.

All reviewers agree that this can be a really interesting contribution to PLOS Genetics but there is some work to do both on clarity of presentation and display items, background information (when available) on the study populations, and specific ideas to further analysis. All of these requests from the reviewers appear very reasonable and should be carefully addressed before resubmission.

If you decide to revise the manuscript for further consideration at PLOS Genetics, please aim to resubmit within the next 60 days, unless it will take extra time to address the concerns of the reviewers, in which case we would appreciate an expected resubmission date by email to plosgenetics@plos.org.

[LINK]

We are sorry that we cannot be more positive about your manuscript at this stage. Please do not hesitate to contact us if you have any concerns or questions.

Yours sincerely,

Mikkel H. Schierup

Associate Editor

PLOS Genetics

Bret Payseur

Section Editor: Evolution

PLOS Genetics

Reviewer's Responses to Questions

**Comments to the Authors:**

Reviewer #1: Dear James and colleagues,

I have enjoyed reading and reviewing your manuscript.

My comments are both brief and minor.

As you note in your manuscript you are working with an elegant system for studying evolutionary responses to different ecological pressures, in this case predation risk.

The system is a great choice for investigating convergent genomic evolution which may underlie repeated phenotypic changes found across the different hp and lp populations.

The sampling scheme and sample size are both well suited to addressing this question.

The introduction contains an excellent overview of the different genomic pathways evolution can take to reach convergent phenotypic outcomes.

The progression through the different analyses is intuitive; first resolving population structure and demographic history, before investigating potential targets of repeated selection across pairs of sister HP and LP populations. We have taken a very similar approach ourselves in a study of coastal and pelagic dolphin populations.

The story is somewhat complex, and I found myself having to re-read sections of the text and scroll back and forth, for example to remind myself of the different abbreviations, river names etc. But I don't have any solution for this and I don't feel this is due to any flaw in the writing or presentation. There just is a level of complexity that requires anyone not already familiar with the study site /system to have to read the manuscript closely. I feel you've made good choices in the level of detail included and the strength of the conclusions drawn. I appreciated the teasing out of the SV underlying the Chr20 peak. Overall, there is no clear functional link between genomic variation and phenotypic variation, but that is often the case with non-model study systems (and could definitely be said about my own papers!). I am sure there will be opportunities to test these associations further perhaps in a lab based setting in future studies. What this study has achieved is identifying some very strong candidate regions for investigation with an equally strong understanding of the underlying evolutionary and demographic history.

I think it is a valuable contribution to our understanding of parallel evolution in nature.

Some minor thoughts as I was reading through the manuscript:

Figure 1D the shapes of different predation regime are not always clear due to dense clustering. Could you use open and filled markers to distinguish HP and LP? and does that make it any clearer?

Lines 164-181 I'm more familiar with the significance of D-statistics being expressed as Z-scores as an indication of the number of standard deviations (estimated from weighted block jackknifing) from 0 under the assumption of a normal distribution of shared allele counts with P1 and P1. I'm not familiar with how the estimates and probability scores from Dsuite are derived. Are you able to provide a few more details.

Lines 366-368. Work by Simon Martin and colleagues has suggested that the D-statistic is not always suitable for inferring specific introgressed regions within the genome. Simon and colleagues have developed alternative approaches, one most recently with Bill Amos using D estimated from multiple samples per population. Do you think Simon's concerns are relevant here? and would one of these alternative methods be worth applying to your data?

Reviewer #2: The authors present an analysis designed to detect genomic convergence in guppies adapted to alternative environments. Rivers in Trinidad have diverse, predator-rich communities in the higher order portions of streams that drain the Northern Range Mountains. The streams often have waterfalls that exclude predators, but not guppies, from the upper reaches of these streams and tributaries. There is an extensive literature that characterizes how guppies have adapted to life without predators. These adaptations include life histories, morphology, swimming performance, male coloration, male courtship behavior and other aspects of behavior. The question is whether or not there is parallel convergence in the evolution of the genome. If so, then this would provide traction for subsequent studies of the genetics of adaptation. The authors make five paired comparisons between guppies from high and low predation environments in five different rivers. Three of these rivers are part of the Caroni river system on the southwest slopes of the mountains, one is from the Oropuche River system on the southeastern slopes and one is from the Madamas River, an independent drainage on the north slopes of the mountains.

The authors scan for portions of the genome that bear evidence of selection with three different types of paired comparisons between high and low predation populations in each of the five rivers and accept the results as positive evidence of selection if at least two of the three methods support that evidence. They find scant evidence for convergence in the form of genetic differences repeated in all five rivers. The one signal, a weak one, is the signature of selection in genes associated with the Cadherin-signaling pathway. The much stronger signal “mapped to a previously unplaced scaffold…at the start of chromosome 20” (lines 278-282). This is the strongest signal in the analysis, but is only evident in the three rivers that were part of the Caroni river system. They present a detailed analysis of the nature of the differences between the HP and LP population in each river, address whether or not it represents an inversion (it appears to not be an inversion, but may instead be a region of restricted crossing over near the centromere) and discuss the genes contained within this region and the ones that carry the strongest signs of selection.

The results are disappointing because of the absence of strong, uniform signals of parallel evolution of the genome, but the analysis is sound. Such convergent genomic evolution has emerged in studies of other classic examples of convergent evolution, most notably the sticklebacks, and the authors discuss why guppies might be different from sticklebacks. This paper includes a thorough analysis of population demography and gene exchange among populations. One feature of guppies that distinguishes them from sticklebacks and offers a potential explanation is that the freshwater populations of sticklebacks are derived from a large, panmictic marine population and appear to have been established by a large number of colonists, so each one begins with a good sampling of the genetic variation from the founder. In guppies, the ancestral populations are genetically distinct from one another and the descendant populations often carry a signature of a genetic bottleneck, meaning that they were established by a small number of founders. Overall, I think that the paper represents a significant step forward in our understanding of the system and merits publication. I do, however, feel that many improvements could be made to the presentation.

General Comments:

1. They must provide better locality data for where the fish were collected from. There are multiple candidates for low predation in all five streams. We need to know which one they chose since they each could have different histories and properties. If nothing else, posterity needs this information if anyone is to follow up on this work. I am especially concerned about the Aripo River since one of the LP tributaries is a site where guppies were deliberately introduced by John Endler in 1976. Their recent origin and having had what was likely an unusually large number of founders would possibly account for the difference between the Aripo and the other two from the Caroni in the results for chromosome 20 (as highlighted on lines 416-418). I might add that this paper could be a good deal more interesting and important if these fish were indeed from the introduction site.

2. Polarization of the direction of evolution: Lines 411-414 state that HP is ancestral and LP is derived plus speculates on why we do not see repeated patterns of molecular convergence across drainages. I think this is an important point that needs to be amplified. First, do their data enable them to address whether or not HP really is ancestral to LP? The fact that HP has higher genetic diversity is not by itself adequate. If we imagine HP as being the composite of separate LP populations distributed through the diverse headwater streams that combine to form HP, then their mixture provides an alternative explanation for the patterns of genetic variation. While I consider this top-down alternative to be unlikely, I think it would add to this paper if their data can be martialed to resolve the ancestral-descendant relationship among populations within each of the five rivers. I don’t know if it is possible, but it would be a great contribution if there were some way to nail down who was ancestral vs. descendant. I strongly favor their interpretation, but it would be great if their data could provide some definitive support for it.

3. Clarity of the figures: I am a genomics novice and I found some of the figures to be uninterpretable. I found better trained colleagues who were able to figure them out for me, but if this paper is to be accessible to a general audience then some additional explanation is required.

Fig. 1e: In my discussions with colleagues it emerged that these figures may have been over-interpreted in the text. The apparent gene flow between Madamas, Oropuche and Aripo may not mean that there has been movement from both Madamas and Oropuche into Aripo. It is also possible that there is movement from Madamas to Oropuche then from Oropuche to Aripo. If this is the case, then you should clarify the extent to which these results lend themselves to alternative interpretations. Elsewhere you say that there is evidence of gene flow between Aripo LP and Oropuche LP, but I see no evidence of that in this figure.

Fig. 1f: I need a more detailed explanation of what the entry in each cell is and a key to what the color coding means, plus an explanation for those cells that are gray and presumably have no entry. My colleagues were able to explain this figure to me but I had no prayer of figuring it out on my own.

Fig. 2a: I again could not understand Fig. 2a on my own but my colleagues were able to explain it. The caption is not adequate for someone without a better genomics background than mine.

Fig. 2c and d: You show some compelling results for chromosome 8 but I saw no more about this in the text. What happened here?

Fig. 3: I had no problem with this one. Fig. 3b seems to report some differentiation between LP and HP in the Oropuche River in this same portion of chromosome 20. What is going on there?

4. Discussion: The discussion is not optimally organized. It is a collection of paragraphs on different topics, all appropriate for a discussion, but all presented in a continuous fashion that is hard to follow. I think it would benefit from being divided up into subheadings:

Paragraph 1 (lines 401-418): an appropriate summary

Paragraphs 2 – 4 (lines 420-477) detail the results for Chromosome 20 and fit well into a single subheading. Some of the paragraphs are overlong and group material that would be better divided into more and simpler paragraphs.

Paragraphs 5-7 (lines 479 – 510) detail population structure, introgression and how structure might explain the scarcity of genomic convergence. You return to this theme in Paragraph 9 on lines 534-542 so perhaps grouping it with these paragraphs under one subheading would make more sense.

Paragraph 8 (lines 512-532) deals with the Cadherin pathway and the different paths to genetic convergence so it seems better to group it under its own subheading.

Additional Comments:

6. Lines 203 – 207 and Fig. S1: The three Caroni River tributaries all have the potential for facilitated gene flow because they are connected and this is reflected in Fig. 1, so why do the steeper mountains that separate these tributaries matter?

7. Lines 214-216: They are no doubt correct in concluding that there is more evidence of gene flow from LP to HP sites within each river, but they also assume, as have others, that HP is ancestral to LP, meaning that migrants from HP founded the LP populations. This is where it would be great if your data could in some way address who is ancestral and who is descendent within each stream.

8. Lines 334-6: I think this statement will be a mystery to anyone who is not very familiar with the guppy literature. It is worth emphasizing that the fish from the Oropuche river system are sufficiently different from those in the Caroni and north slope to have been named a different species by one investigator. But isn’t this pattern consistent with the CL haplotype having arisen within the common ancestor of the Caroni river populations then moved into the Oropuche via gene flow?

9. Lines 391-392: Is “lost” the correct word? Might “absence” be better since it implies that the CL haplotype could also have originated in the common ancestor of the Caroni River populations? I either have missed something in your explanation of these results or you are not considering this alternative explanation.

Reviewer #3: Review for: “Drainage-structuring of ancestral variation and a common functional pathway shape limited genomic convergence in natural high- and low-predation guppies”

General comments:

Instances of convergent and parallel evolution facilitate the discovery of common rules of organismal diversification. In this study, Whiting et al. explore genetic basis underlying convergent evolution of Trinidadian guppies adapted to high- and low- predation (HP and LP) pressures. The authors sampled ~15 HP and ~15 LP individuals from five rivers and obtained whole genome data at ~10x coverage. Overall, I think the core questions are interesting. The study system, the sampling scheme, and the genomic dataset are well suited to address the questions. In terms of genetic convergence, the authors find a haplotype on chr20 and genes involved in cadherin signalling appear to be involved to some degree, but not much else, and nothing at the level of individual genes or variants. The authors conclude that this is because of a lack of shared genetic variation between LP populations. The writing is generally clear and possible to follow, but not always – it would benefit from some improvements – on occasion, the expressions and terminology used sound somewhat clumsy, and there is too much repetitiveness. I have two major concerns and a number of comments and suggestions for the authors to consider before publication.

Major comments/concerns:

First, I find that the paper lacks background information. This limits my ability to understand if the methods employed are appropriate and also how the paper fits into the broader context of other empirical studies of convergent evolution.

Second, the conclusion that the paucity of genetic convergence is because of a lack of shared genetic variation is speculative. The authors did not test this in any way; all they have is indirect evidence in the form of inferred bottlenecks.

I am going to expand on both of these points with specific comments/suggestions .

Specific comments/suggestions:

In the introduction, and in fact anywhere in this paper, I am missing information about the timeframe of evolution of these convergent phenotypes. Is anything known about when the waterfalls separating the HP and LP populations formed? At least a ballpark figure – 50 years, 5000 years, 5 million? And how much does this differ between the rivers? The ages of the rivers? Or some inference from the genetic data?

Another thing that is missing is some idea as to how far has the adaptation proceeded in the five rivers. If the relative ages of the waterfalls are different, perhaps the LP populations are adapted to different degrees? Also, even if the ages are the same, are the phenotypes always the same? Have these specific populations been phenotyped previously? Could some phenotyping (e.g. colour?) be done for the current study?

Lines 135-142: “PC2 (19.8%) separated out Caroni rivers, highlighting population structure within this drainage is stronger than structure between Madamas and Oropouche, despite these rivers being in separate drainages…….”

- This may also be simply a sample size effect – I suggest you try to verify this by doing a PCA including only two Caroni rivers and Madamas and Oropouche.The same for fineSTRUCTURE. I suspect that your conclusions may change.

Lines 164-181 and Figure 1E:

- I suggest that you plot and cite in text the f4-ratio statistics, not the D. The f4-ratio estimates the admixture proportion, and thus, unlike D, it is comparable between studies in a biologically meaningful way as a measure of gene-flow strength. On the other hand, the magnitude of D depends on the amount of incomplete lineage sorting.

Figure 1B,C, and F –

All three panels lack scale bars. For 1B, perhaps a proper phylogeny would be more appropriate than the fineSTRUCTURE clustering? And for 1F, I wonder why the authors did not use the unfolded site frequency spectrum, which would be more informative and perform better in fastsimCoal2. The authors did add an outgroup to the VCF for the Dsuite analyses, so they could use it to polarise ancestral-derived alleles.

Maybe most importantly, given the conclusion that of a lack of genetic convergence is due to a lack of shared genetic variation, it would be interesting to know how much shared genetic variation there is among the five LP populations. Maybe create a 2d site frequency spectrum for pairs of LP populations?

The selection tests raise two thoughts:

a) The methods – both XtX and AFD reflect allele frequency shifts in slightly different ways. XP-EHH is different, reflecting recent selection. Since the authors state “Putatively selected windows were identified if they were detected as outliers by at least two approaches” – I wonder how often this simply reflects an agreement between the two allele-frequency-based approaches?

b) With fixed 10kb windows, how much are the results driven by stochasticity in the number of SNPs per window? The windows with fewer SNPs will have greater variance and present outliers more often. Perhaps trying to apply the analysis with windows that have equal numbers of SNPs would be interesting?

I liked the permutation tests for pathway enrichment.

Language, typos etc.:

Lines 44-45: “convergent phenotypes are often underwritten by convergent changes at the genetic level”

Underwritten sounds a bit like we are reading about insurance or finance. How about “convergent phenotypes are often encoded by …. “, or something like that?

Lines 69-70: “Given the range of genetic convergence observed across empirical studies, the importance of different contingencies have emerged.”

- How about: “Given the divergence in modes and levels of genetic convergence observed across empirical studies, different contingent factors have emerged as important.”

Line 72-73: “Limitations in this map are expected to upwardly bias reuse of the same genes or mutations, but redundancy can allow for the evolution of different genes in shared functional pathways.”

- This takes some deciphering. Why “limitations in this map”? Do you mean “limitations in the redundancy of this map”? Why “bias”? Why “allow for”?

- Maybe you mean something like this: “Simple, e.g. one-to-one, mapping is expected to result in reuse of the same genes or even mutations, while redundancy can lead to convergent phenotypes for example by selection on different genes in shared functional pathways.”

Lines 80-83: “whereby consistencies in the multidimensional fitness landscape channel adaptation along conserved paths. These latter two limiting factors may also explain why genetic convergence can vary for the same traits in the same species in global comparison.”

- I don’t understand any of the logic here – if consistencies in fitness landscape channel adaptation, why does genetic convergence vary?

Many similar cases follow, but in general the paper is possible to follow.

Much of the discussion repeats the results. I suggest cutting down the repetition…

**Have all data underlying the figures and results presented in the manuscript been provided?**

Reviewer #1: None

Reviewer #2: Yes

Reviewer #3: None

PLOS authors have the option to publish the peer review history of their article (what does this mean?). If published, this will include your full peer review and any attached files.

Reviewer #1: No

Reviewer #2: No

Reviewer #3: No

---

## [Decision Letter · Decision Letter 1]

27 Apr 2021

Dear Dr Whiting,

We are pleased to inform you that your manuscript entitled "Drainage-structuring of ancestral variation and a common functional pathway shape limited genomic convergence in natural high- and low-predation guppies" has been editorially accepted for publication in PLOS Genetics. Congratulations!

Yours sincerely,

Mikkel H. Schierup

Associate Editor

PLOS Genetics

Bret Payseur

Section Editor: Evolution

PLOS Genetics

Comments from the reviewers (if applicable):

Reviewer's Responses to Questions

**Comments to the Authors:**

Reviewer #1: I have read the revised manuscript and the author's response to my comments.

They have done a good job in responding and acting upon all my feedback.

I have no further comments and recommend acceptance.

Reviewer #2: I am fully satisfied with the revisions to the paper. I (rev. 2) and the other reviewers placed large demands on the authors to revise what was already a well written paper. They have thoroughly responded to all of our requests in terms of performing new analyses, modifying the figures appropriately, and revising the text. I have no further requests for revision.

Reviewer #3: Review for 1st revision of: “Drainage-structuring of ancestral variation and a common functional pathway shape limited genomic convergence in natural high- and low-predation guppies”

I was happy to see that this version of the manuscript is much improved. Well done to the authors. Most importantly, the authors have addressed my scientific concerns satisfactorily.

There are a few remaining minor comments:

- There is still too much repetitiveness overall and the discussion especially feels too long.

- Line 51: “different mutations in the same genes” -- Does the mutation need to be “in the gene”? Maybe the authors mean “different mutations affecting the same genes”?

- Line 177: “and so most likely represent extensive incomplete lineage sorting”. Incomplete lineage sorting alone, however extensive, cannot lead to significantly elevated D statistics. The true reason for these D results is unclear, but the fact that the corresponding f4-ratios are low suggests that they are not biologically very important (i.e. the proportion of the genome affected is very low). In any case, I suggest you remove the quoted part of the sequence, as this is definitely incorrect.

I could probably find more smaller issues, and there is still some clumsy writing, but I trust the authors will proofread it carefully and deal with that in the next round.

**Have all data underlying the figures and results presented in the manuscript been provided?**

Reviewer #1: Yes

Reviewer #2: Yes

Reviewer #3: Yes

PLOS authors have the option to publish the peer review history of their article (what does this mean?). If published, this will include your full peer review and any attached files.

Reviewer #1: **Yes: **Andy Foote

Reviewer #2: No

Reviewer #3: **Yes: **MIlan Malinsky

**Data Deposition**

http://datadryad.org/submit?journalID=pgenetics&manu=PGENETICS-D-20-01641R1

**Press Queries**

---

## [Editor Report · Acceptance letter]

20 May 2021

PGENETICS-D-20-01641R1 

Drainage-structuring of ancestral variation and a common functional pathway shape limited genomic convergence in natural high- and low-predation guppies 

Dear Dr Whiting, 

We are pleased to inform you that your manuscript entitled "Drainage-structuring of ancestral variation and a common functional pathway shape limited genomic convergence in natural high- and low-predation guppies" has been formally accepted for publication in PLOS Genetics! Your manuscript is now with our production department and you will be notified of the publication date in due course.

With kind regards,

Agota Szep

PLOS Genetics

On behalf of:
